# COVID-19 Vaccine Hesitancy—A Scoping Review of Literature in High-Income Countries

**DOI:** 10.3390/vaccines9080900

**Published:** 2021-08-13

**Authors:** Junjie Aw, Jun Jie Benjamin Seng, Sharna Si Ying Seah, Lian Leng Low

**Affiliations:** 1Outram Community Hospital, SingHealth Community Hospitals, 10 Hospital Boulevard, Singapore 168582, Singapore; sharna.seah.s.y@singhealthch.com.sg (S.S.Y.S.); low.lian.leng@singhealth.com.sg (L.L.L.); 2MOH Holdings Pte Ltd., 1 Maritime Square, Singapore 099253, Singapore; benjamin.seng@u.duke.nus.edu; 3Department of Family Medicine and Continuing Care, Singapore General Hospital, Singapore, Outram Rd, Singapore 169608, Singapore; 4SingHealth Duke-NUS Family Medicine Academic Clinical Program, Outram Rd, Singapore 169608, Singapore; 5SingHealth Regional Health System PULSES Centre, Singapore Health Services, Outram Rd, Singapore 169608, Singapore

**Keywords:** scoping review, coronavirus disease-19 (COVID-19), COVID-19 pandemic, SARS-CoV-2 infection, 2019 novel coronavirus disease, vaccines, COVID-19 vaccines, vaccine hesitancy, vaccine acceptance

## Abstract

Vaccine hesitancy forms a critical barrier to the uptake of COVID-19 vaccine in high-income countries or regions. This review aims to summarize rates of COVID-19 hesitancy and its determinants in high-income countries or regions. A scoping review was conducted in Medline^®^, Embase^®^, CINAHL^®^, and Scopus^®^ and was reported in accordance with the PRISMA-SCr checklist. The search was current as of March 2021. Studies which evaluated COVID-19 vaccine hesitancy and its determinants in high-income countries (US$12,536 or more GNI per capita in 2019) were included. Studies conducted in low, lower-middle, and upper-middle income countries or regions were excluded. Factors associated with vaccine hesitancy were grouped into four themes (vaccine specific, individual, group, or contextual related factors). Of 2237 articles retrieved, 97 articles were included in this review. Most studies were conducted in U.S. (*n* = 39) and Italy (*n* = 9). The rates of vaccine hesitancy across high-income countries or regions ranged from 7–77.9%. 46 studies (47.4%) had rates of 30% and more. Younger age, females, not being of white ethnicity and lower education were common contextual factors associated with increased vaccine hesitancy. Lack of recent history of influenza vaccination, lower self-perceived risk of contracting COVID-19, lesser fear of COVID-19, believing that COVID-19 is not severe and not having chronic medical conditions were most frequently studied individual/group factors associated with increased vaccine hesitancy. Common vaccine-specific factors associated with increased vaccine hesitancy included beliefs that vaccine are not safe/effective and increased concerns about rapid development of COVID-19 vaccines. Given the heterogeneity in vaccine hesitancy definitions used across studies, there is a need for standardization in its assessment. This review has summarized COVID-19 vaccine hesitancy determinants that national policymakers can use when formulating health policies related to COVID-19 vaccination.

## 1. Introduction

Since its first reported case in December 2019, the coronavirus-2019 (COVID-19) pandemic has culminated in nearly 179 million infections and 3.88 million deaths globally as of 24 June 2021 [1]. Lockdowns, social distancing measures, and movement restrictions were implemented as a result to abate the spread of infection worldwide [2]. The aftermath of the pandemic has negatively affected global economies. For example, the Internal Labor Organization has estimated 25 million jobs to be lost and the United Nations World Tourism Organization has estimated a loss of US$80 billion dollars in international travel receipts internationally in 2020 [3,4].

Vaccination forms a critical pillar in the road to recovery from the COVID-19 pandemic [5]. Notably, vaccine candidates with promising results received expeditious emergency use authorization by drug authorities. Despite quick and concerted vaccination programs implemented by governments globally, such efforts have been hampered by vaccine hesitancy. Vaccine hesitancy was identified by the World Health Organization as one of the 10 threats to global health in 2019. It is defined as the “delay in acceptance or refusal of vaccination despite availability of vaccination services” by the Strategic Advisory Group of Experts on immunization and involves a complex interaction of time, place, context, and vaccine specific factors [6].

Among non-high income countries or regions, results from the 2018 Wellcome Global monitor survey showed that vaccines were widely accepted [7,8], in contrast to high-income countries (defined by World Bank as countries having a 2019 Gross National Income (GNI) per capita of US$12,536 and more) [9]. A recent study by Arce et al. showed that the average willingness to take COVID-19 vaccine was higher in the populace from non-high income countries or regions such as Nepal (97%) as compared to those in high-income countries or regions such as United States (6%) [10]. Similar findings were noted in a study by Sallam et al. [11].

In view of the importance of COVID-19 vaccine hesitancy, we aim to perform a scoping review to evaluate COVID-19 vaccine hesitancy and its determinants among high-income countries or regions. We hope that our results will aid healthcare administrators and policymakers in understanding COVID-19 vaccine hesitancy determinants in high-income countries or regions. This will, in turn, aid and facilitate the planning of vaccination campaigns to enhance uptake of COVID-19 vaccinations.

## 2. Methodology

We conducted a scoping review on studies which evaluated COVID-19 vaccine hesitancy in high-income countries or regions. This review was reported using the Preferred Reporting Items for Systematic review and Meta-Analysis extension for Scoping Reviews (PRISMA-ScR) checklist [12].

### 2.1. Protocol and Registration

The protocol for this scoping review is registered on 11 April 2021 on Open Science Framework (Available online: https://osf.io/3n7yv (accessed on 11 April 2021)).

### 2.2. Eligibility Criteria and Information Sources

A literature search was performed in four major literature databases which were namely: Medline^®^, Embase^®^, CINAHL^®^, and Scopus^®^. Full-text articles in English language which evaluated COVID-19 hesitancy rates and the associated determinants in high-income countries or regions were included. Study designs in this review included randomized controlled trials, observational studies, cross-sectional studies, cohort studies and qualitative studies. We excluded studies that were performed in non-high-income countries (GNI per capita < US$12,535). Commentaries, editorials, letters and correspondences without original data as well as irrelevant systematic reviews and meta-analyses were excluded. The search period for the review spanned between December 2019 and March 2021. Institutional review board approval was exempted for this review as it did not involve human subjects. 

### 2.3. Search Strategy

The search strategy comprised of two main themes which were COVID-19 vaccine and vaccine hesitancy. The search strategy used was adapted from prior systematic reviews which evaluated vaccine hesitancy related to other vaccines [13,14,15]. The full details of the search strategy is available from Appendix A.

### 2.4. Selection of Sources of Evidence, Data Charting Process, and Data Items

Citations retrieved from the four databases were exported into Endnote Software Online (Clarivate Analytics, Philadelphia, PA, USA). Duplicated citations were removed prior to screening of articles. Two independent reviewers (J Aw and JJB Seng performed the initial pilot exercise for the screening of the first 200 records (based on title and abstract). Thereafter, the titles and abstracts of all retrieved articles from the four databases were screened by the same reviewers independently. The full-text articles of potentially relevant articles were evaluated prior to inclusion in this review. All disagreements in the inclusion phase of the review were discussed to reach a consensus. For discrepancies which could not be resolved between the two reviewers, arbitration was made with a third independent reviewer (SYS SEAH).

To chart data from the included articles, a standardized Microsoft Excel data collection sheet was used. This information included the name of author, title of study, publication year, sample size, study design and methodology, characteristics of patient population, tools used to evaluate COVID-19 vaccine hesitancy, reported hesitancy rates, and determinants associated with increased COVID-19 vaccine hesitancy.

### 2.5. Critical Appraisal of Individual Sources of Evidence

The risk of bias appraisal for included studies was not performed as this was not the objective of this scoping review. 

### 2.6. Summary and Synthesis of Results

Descriptive statistics were used to summarize the characteristics of studies included in this review. Vaccine hesitancy rate was reported from individual study according to the definition described in each study. In studies which reported only vaccine acceptance rates, vaccine hesitancy rates were computed using the formula: [100 (%)—vaccine acceptance rates (%)]. In cohort studies which reported longitudinal rates of vaccine hesitancy, the mean vaccine hesitancy rates were extracted. Other variables collected included the study design and methodology, characteristics of participants and determinants of COVID-19 vaccine hesitancy. Graphical charts and tables were used to present the results.

There is no widely accepted definition for cut-off with regards to a high vaccine hesitancy rate. Assuming COVID-19 vaccines can stop transmissibility and that COVID-19 has a R_0_ of 2–3.5, a 60–70% vaccination uptake is estimated for herd immunity [16]. We therefore define high vaccine hesitancy as 30 or more percent in this review.

A narrative summary of factors associated with increased COVID-19 vaccine hesitancy was presented. The determinants of vaccine related hesitancy were grouped into three main categories: contextual influences, individual/group influences, and vaccine/vaccination specific issues, as proposed by the Strategic Advisory Group of Experts (SAGE) on immunization [6,17]. These determinants were reported as per described in individual studies.

A framework diagram was used to summarize the most frequently studied determinants of COVID-19 vaccine hesitancy in high-income countries or regions.

### 2.7. Data Availability Statement

Data analyzed in the study is included in the published article and Appendix A.

## 3. Results

A flowchart for inclusion of articles in this review is illustrated and of 2237 citations retrieved, a total of 97 articles were included based on the inclusion criteria (Figure 1). The percentage of agreement between the reviewers during the inclusion was 90.7%.

### 3.1. Characteristics of Included Studies

In the summary table of the characteristics of included studies, approximately half of the included studies were conducted in Year 2021 (50.5%) while the other half were performed in 2020 (49.5%) (Table 1). Most of the studies were conducted in North America (43.3%) and Europe (34.0%). Four studies involved cross-continent collaborations. Of note, the two countries with the highest number of studies were U.S. (*n* = 39, 40.2%) and Italy (*n* = 9, 9.3%). Cross-sectional study design (*n* = 75, 77.3%) and online survey methodology (*n* = 87, 89.7%) were most frequently described in studies included. Further details of included studies are available in Appendix A. 

### 3.2. Study Population

Across the populations studied, most studies evaluated vaccine hesitancy rates among the general public (*n* = 71, 73.2%) and healthcare workers (*n* = 13, 13.4%). Other studied populations included university students/staff (*n* = 5, 5.2%) and patients (*n* = 4, 4.1%). Among the included studies, only 10 studies (10.3%) evaluated participants’ hesitancy towards COVID-19 vaccination for their children [15,18,19,20,21,22,23,24,25].

### 3.3. Vaccine Hesitancy Rates across Studies

Figure 2 shows a bar chart illustrating vaccine hesitancy rates and number of studies done in high income countries. Among the 97 studies included, 46 studies had vaccine hesitancy of 30% and more (Table 1). Among the four continents exploring vaccine hesitancy in high income countries, Asia had the highest proportion of studies with vaccine hesitancy of 30% or more [*n* = 8 (72.7%)] while North America ranked second [*n* = 25 (59.5%)]. Studies conducted in Europe and Oceania had a lower proportion of studies with vaccine hesitancy 30% or more. Individually, vaccine hesitancy rates were highest in UAE (77.9%), U.S. (66.8%), Hong Kong (60%), and Italy (59.9%). In contrast, the vaccine hesitancy rates were lowest in Canada (7%) and Saudi Arabia (7%).

Reasons for vaccine hesitancy/acceptance were explored in 21 studies through open ended questions but only 7 (7.2%) attempted to describe methods on thematic analysis. Of these, only four are qualitative studies with well described methodologies [23,25,26,27]. Cross-sectional studies (*n* = 75, 77.3%) were the most frequent study designs while online surveys (83, 85.6%) were the most frequently used methods ^a^.

Pertaining to the definition of vaccine hesitancy used to derive its proportion, just slightly over half of the studies (51.5%) conformed to SAGE working group definition of vaccine hesitancy ^a^. ^a^ Detailed data available in Appendix A.

### 3.4. Determinants of Vaccine Hesitancy

#### 3.4.1. Contextual Related Factors

A total of 25 themes were identified and grouped under the eight sub-categories in “Contextual determinants of vaccine hesitancy” (Table 2). “Sociodemographic related variables”, “policies/politics related factors”, and “communications and media environment related factors” were most frequently studied themes.

Among the sociodemographic variables, being females (*n* = 37), [15,20,24,33,35,36,37,40,41,44,46,48,50,51,52,53,54,56,57,58,59,60,61,62,63,64,65,66,67,68,69,70,71,72,73,74,75] having a younger age (*n* = 31) [32,35,36,37,41,46,49,53,54,56,60,63,65,66,67,68,70,71,72,73,76,78,79,83,84,85,86,87,88,89] being of non-White ethnicity (*n* = 24), [23,31,32,37,40,41,45,47,49,51,52,54,64,65,68,70,71,72,78,86,87,89,93] having a lower education (*n* = 19) [32,34,36,41,49,54,65,66,68,69,71,77,82,85,86,89,95,96,97] and a lower income level (*n* = 13) [23,30,40,44,48,49,51,52,53,54,82,89,97] were associated with vaccine hesitancy.

With regards to policies and politics related factors, political inclination towards non-democrats in the U.S. (*n* = 8) [33,40,41,43,44,45,46,47] and non-liberals (*n* = 8) [40,46,48,49,50,51,52,53] were associated with vaccine hesitancy.

For communications and media environment factors, the use of social media or internet as a main source of information (*n* = 6) [28,29,30,31,32,33] and the lack of widely accessible information on COVID-19 vaccination (*n* = 5) [29,35,36,37,38] were associated with vaccine hesitancy.

Other notable factors associated with vaccine hesitancy included healthcare workers in non-clinical roles (compared to those in clinical roles) (*n* = 7), [15,61,64,65,70,73,87] increased religiosity (*n* = 5), [31,37,40,56,57] residing in rural areas (*n* = 5), [46,65,68,86,100] reduced trust in government and pharmaceutical industry (*n* = 7) [20,37,41,56,79,92,102] and increased passage of time in a pandemic (*n* = 5) [34,68,71,75,94].

Two studies found an increased vaccine hesitancy mainly in nursing staff among healthcare workers with clinical fronting roles [67,72].

#### 3.4.2. Group/Individual Related Factors

A total of seven sub-categories of factors with 22 themes were identified for the “Individual/group determinants of vaccine hesitancy” (Table 3). “Beliefs, attitudes about health and prevention”, “past experiences with vaccinations”, and “health-system and providers—trust and personal experience” were most well-studied.

A lesser fear for health or worry about COVID-19 (*n* = 16), [32,33,35,37,38,40,49,53,54,59,61,67,68,79,81,90] a perception of lower risk of contracting COVID-19 (*n* = 15), [28,43,44,49,52,54,59,61,63,81,94,96,98,104,105] believing that COVID-19 is not severe (*n* = 12), [22,36,41,49,52,58,63,64,81,92,98,105] having lesser trust in healthcare system (*n* = 11) [22,30,32,50,62,68,73,92,106,107,108] and believing that vaccination is unimportant or non-beneficial (*n* = 12) [20,31,37,43,54,68,78,90,91,92,102,105] were most frequently studied associations with increased vaccine hesitancy.

Previous influenza vaccination was the most common determinant associated with lower vaccine hesitancy (*n* = 28) [15,19,20,22,25,33,35,36,37,41,46,47,49,50,58,61,65,66,68,73,74,75,78,79,86,89,90,95,104].

#### 3.4.3. Vaccine Related Factors

A total of 10 themes were identified and grouped under the original eight sub-categories for “Vaccine related determinants” of vaccine hesitancy (Table 4). Among these, factors related to “risk and benefits of the vaccine” and “introduction of new vaccine/formulation” were the most studied subcategories.

The most studied determinants associated with increased vaccine hesitancy included beliefs that COVID-19 vaccines are unsafe or ineffective (*n* = 24) [20,22,29,35,36,37,40,41,43,50,52,54,55,63,68,73,88,90,94,102,103,108,110,111] and concerns related to the rapid development of vaccine and/or its mechanism of action (*n* = 9) [20,22,24,41,43,50,54,108,110].

Other notable factors associated with increased vaccine hesitancy included presence of perceived barriers to accessibility of vaccine (*n* = 4), [36,92,97,111] lack of advocacy for vaccination by attending physicians (*n* = 3) [49,52,88] and multidose nature for vaccination schedule (*n* = 2) [88,110]. Evidence linking concerns about cost for vaccination (*n* = 4) were mixed [43,90,97,104].

## 4. Discussion

This review has highlighted a few salient points and some research gaps.

Firstly, it showed that despite the variable rates of vaccine hesitancy across high-income countries or regions, nearly half of studies reported vaccine hesitancy of 30% or more. Our review discovered that only slightly more than half of the studies conducted on COVID-19 vaccine hesitancy conformed to the SAGE proposed definition. In those studies which did not conform, participants who expressed being “unsure” instead of rejecting the vaccine were excluded in the hesitancy rate, leading to a potential falsely reassuring low hesitancy rate.

Studies conducted among high income regions across four continents revealed a high proportion of studies with high vaccine hesitancy mostly in Asia and North America. Countries with the highest vaccine hesitancy rates included UAE, U.S., Hong Kong, and Italy. Compared to low-income countries or regions, the current vaccine hesitancy rates in high income countries or regions are worrisome.

The varying vaccine hesitancy rates across countries or regions are complex and may partly be attributed to differences in ideological beliefs, demographics, and context specific factors, as seen for other vaccinations. For example, vaccine hesitancy appears to have a lesser impact on general vaccine uptake rates in lower-middle income countries or regions and affects lower socioeconomic status individuals to a greater extent [112]. The reasons have been linked to disparities in access, cost, and awareness of vaccines [113]. In contrast, individuals residing in more affluent countries or regions tend to be more vaccine hesitant due to concerns related to the safety of vaccines [114]. This is especially so in the current choices of vaccines made with newer technology which raised doubts and long term safety concerns [115].

The global vaccination census showed that the share of population fully vaccinated against COVID-19 stood at 18.3% in high income countries or regions as of May 2021 [116]. Of note, the proportions of population fully vaccinated against COVID-19 in U.S., Italy, Hong Kong, and UAE were at 36%, 13.7%, 10%, and 38.8% as of May 2021 respectively, reflecting our review results with only UAE bucking the trend [116]. In spite of this, there were flattening of the epidemic curves from February onwards in the U.S. and Israel after the commencement of vaccination exercises, reinforcing the importance of vaccination [1].

The second point our study highlighted was to summarize determinants of COVID-19 vaccine hesitancy that were the frequently studied (Figure 3). Females, being younger, having a non-Whites ethnicity and having a lower socioeconomic status (e.g., lower education or income levels) were common demographics identified with COVID-19 vaccine hesitancy (Table 2).

Literature discussing higher vaccine hesitancy in females suggested the underlying reasons attributable to lower perceived risk of COVID-19, higher beliefs in conspiracy related theories about the pandemic compared to their male counterparts [117] and concerns about safety of vaccination in pregnancy and breastfeeding [118].

The association between younger individuals and COVID-19 vaccine hesitancy may be a result of increased public health focus on vaccinating the elderly (due to their risk for severe COVID-19 outcomes) and the lack of outreach on COVID-19 vaccination in social media platforms which they commonly frequent [119].

With respect to ethnicity, Blacks have been shown to have increased mistrust in COVID-19 vaccination with possible reasons due to racism, discrimination and mistreatment within the healthcare systems [120]. We should extrapolate and observe for similar associations to all at-risk populations so that governments and healthcare professionals alike can assess and direct efforts on improving COVID-19 vaccination uptake rates.

Our review also discovered that users of social media/internet as a primary source of COVID-19 related information were more prone to increased vaccine hesitancy. With the advent of infodemic on these non-traditional media platforms, innovations on ways to deliver accurate and timely health information by traditional and non-traditional platforms have become incredibly important. Employing active strategies such as pre-emptive cognitive inoculation techniques and pre-bunking techniques have also been suggested to tackle misinformation [121]. Clear and honest communications form an important bridge between building public trust and reinforcing positive health behaviors or compliance with COVID-19 vaccination [122].

In addition, the other determinants previously mentioned should also be systematically addressed. While it is not within the scope of this review, the way different themes are being measured such as knowledge about COVID-19 disease and vaccination, is an important area of research impacting on the study of vaccine hesitancy across different populations. Our review noted that most studies used self-designed instruments in the evaluation of COVID-19 vaccination knowledge which limits cross-comparison of knowledge levels across populations. Future research should consider developing a standardized instrument for the assessment of knowledge of COVID-19 vaccine and disease which can potentially be adapted for future pandemics.

Hopefully, the summary of these determinants will allow policymakers at the national level to deep dive into local context and conduct multi-pronged, multi-tiered studies coupled with interventions to overcome vaccine hesitancy in high income countries.

With the ongoing vaccination drive globally and evolving landscape for COVID-19, it remains premature to conclude the real-world impact of vaccine hesitancy on the true uptake of COVID-19 vaccination. Uptake can be confounded by logistic and administrative challenges in vaccine deployment, vaccine production capacity issues from manufacturers, affordability of vaccines and global allocation of vaccines in the context of limited supplies [123]. This was observed in U.A.E. which had one of the highest percentages of population fully vaccinated for COVID-19 (39.3%) globally in May 2021 despite a reported high vaccine hesitancy [124]. In contrast, the percentage of fully vaccinated population in Canada, which had the lowest vaccine hesitancy, was only 3.3% in May 2021 [124].

Nonetheless, vaccine hesitancy studies will continue to provide insights into possible future directions to drive vaccination efforts. In planning vaccination programs, two considerations related to COVID-19 vaccination are important moving forward. Firstly, if COVID-19 vaccinations can stop transmissibility of COVID-19, at least 60–70% of population needs to be vaccinated [125]. Secondly, in the scenario where COVID-19 vaccines reduce only disease severity but not transmissibility, identifying targeted groups for priority vaccination will become the de facto strategy. Studying vaccine hesitancy across patient subgroups who have the highest mortality and morbidity related to COVID-19 infection will be of paramount importance.

Several research gaps related to COVID-19 related vaccine hesitancy were identified in this review. Currently, there are limited studies which have evaluated longitudinal changes in COVID-19 related vaccine hesitancy. COVID-19 vaccine hesitancy may fluctuate or even increase due to fatigue with lockdown and preventive measures, or secondary to increased complacency coupled with reduced risk perceptions with a long duration of pandemic [126]. Future studies may want to consider evaluating the variation in vaccine hesitancy at different timepoints in the COVID-19 pandemic, given its continued waves of outbreaks in different countries or regions currently. There is also paucity of data related to COVID-19 vaccine hesitancy among pediatric groups as well as a lack of assessment of parental concerns of COVID-19 vaccinations in children. As data from studies evaluating the safety and efficacy of COVID-19 vaccination among children emerges soon, it is an important research area to explore. In addition, this review had noted a dismal number of qualitative studies on COVID-19 vaccine hesitancy. Qualitative studies often enable new themes to be identified which is important for comprehensiveness [127]. Albeit challenges abound in conducting qualitative research due to current climate of social distancing measures and lockdowns, some recommended ways to overcome them include use of digital text communications, video diaries and photovoice, where physical interaction can be minimized [128].

This review is not without its limitations. Firstly, the determinants of vaccine hesitancy listed in this review were factors identified from most studies which employed online surveys predominantly. While this was inevitable given the lockdowns and travel restrictions imposed during the COVID-19 pandemic, population groups with limited access to the internet such as older adults, may not be comprehensively captured.

The findings from online studies may be influenced by self-selection bias, survey fraud, and inability of respondents to seek clarity on questions [129]. Among the included studies only a small proportion of online survey studies reported their findings according to the CHERRIES checklist of internet E-surveys [130]. Future studies should consider adopting this checklist to enhance the scientific rigor of their findings. Moreover, among the included studies, we had noticed a significant number of studies not reporting the education level of participants recruited (*n* = 38; 39.2%, data available in Appendix A). A higher level of education in the participants is associated with the possession of correct information on COVID-19 and less susceptibility to misinformation [131].

Secondly, the grey literature as well as literature from pre-print servers were not searched in this review. Future systematic reviews which seek to evaluate vaccine hesitancy among specific populations or perform an updated review should consider searching these resources to improve the comprehensiveness of the search.

Thirdly, among the 57 themes of vaccine hesitancy found in the systematic review, 26 (45.6%) themes had fewer than five studies. The percentages in Table 2, Table 3 and Table 4 with themes having fewer than five studies were *n* = 9 (36%), 40.9% *n* = 9 (40.9%), and *n* = 8 (80%) respectively. A possible insufficient exploration of a theme in the included studies has to be taken into consideration while interpreting and contextualizing the results to individual country.

We would also like to point out a preponderance of studies done in the U.S. exploring the two sub-categories on “policies/politics” and “influential leaders, gate-keepers and anti or pro-vaccination lobbies”. Due to geopolitical differences, generalizability of these themes may be limited.

Lastly, assessments of methodological quality of the included studies, presentations of strength of statistical associations with vaccine hesitancy and meta-analyses of the vaccine hesitancy rates were not performed as these were not the primary aims of this scoping review. Moreover, the heterogeneity in the definition and assessment of vaccine hesitancy in different studies would not have allowed a meaningful meta-analysis. Researchers who are planning to investigate COVID-19 vaccine hesitancy may want to consider adopting the standardized definition of vaccine hesitancy from SAGE workgroup in future studies. This will facilitate and enable future systematic reviews and meta-analyses to evaluate the variation in vaccine hesitancy rates across countries or regions, as well as the temporal variation in vaccine related hesitancy.

## 5. Conclusions

Overall, COVID-19 vaccine hesitancy remains a highly prevalent problem in high income countries or regions. Individuals who were younger, females, non-Whites, and have a lower education or income levels, were more prone to vaccine hesitancy. Trust at different systems levels seem to play an important role in modifying vaccine hesitancy as well. Other commonly studied factors associated with vaccine hesitancy included a history of not receiving influenza vaccination, a lower self-perceived risk of contracting COVID-19, a lesser fear for health outcomes or COVID-19, not believing in the severity of COVID-19, having concerns about the rapid development of COVID-19 vaccines as well as disbeliefs in the safety and effectiveness of the vaccines. Healthcare administrators need to be cognizant of these determinants of vaccine hesitancy when formulating policies related to COVID-19 vaccination and public health messages.

## Figures and Tables

**Figure 1 vaccines-09-00900-f001:**
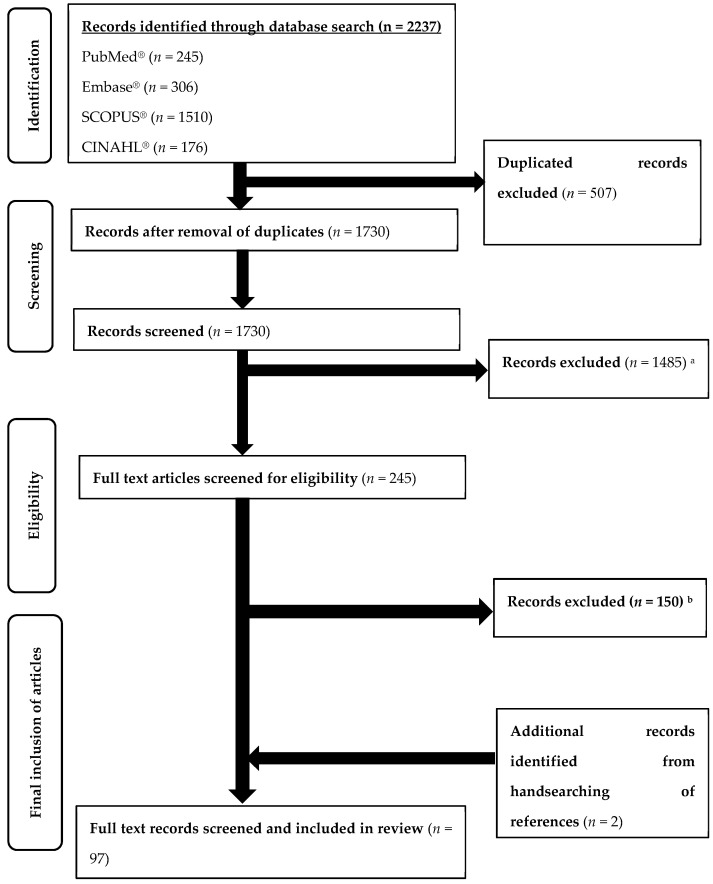
Flowchart for retrieval of articles. ^a^ Records excluded: Did not evaluate COVID-19 related vaccine hesitancy (*n* = 1425); Evaluated vaccine hesitancy in non-high-income countries (*n* = 34); non-English articles (*n* = 23); irrelevant systematic reviews/meta-analyses (*n* = 3); ^b^ Records excluded: Studies included countries other than high-income countries (*n* = 14); studies are editorials, commentaries, news article and/or opinions without original data (*n* = 132); Study looked at willingness of guardians enrolling children in vaccine trials (*n* = 1); studies retracted (*n* = 3).

**Figure 2 vaccines-09-00900-f002:**
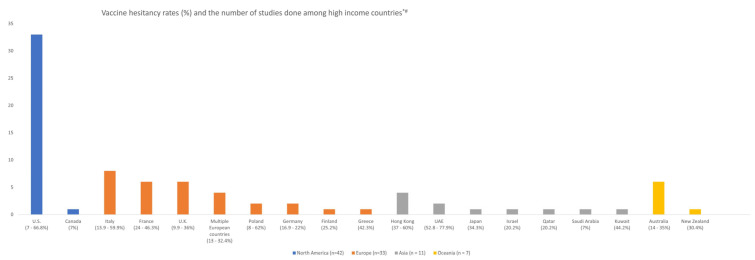
Studies on vaccine hesitancy and its corresponding rates among high income countries. * Only studies with reported vaccine hesitancy rates were included in Figure 2. # 4 studies performed in multi-continents not included to prevent double counting. Abbreviations: U.S.—United States; UAE—United Arab Emirates.

**Figure 3 vaccines-09-00900-f003:**
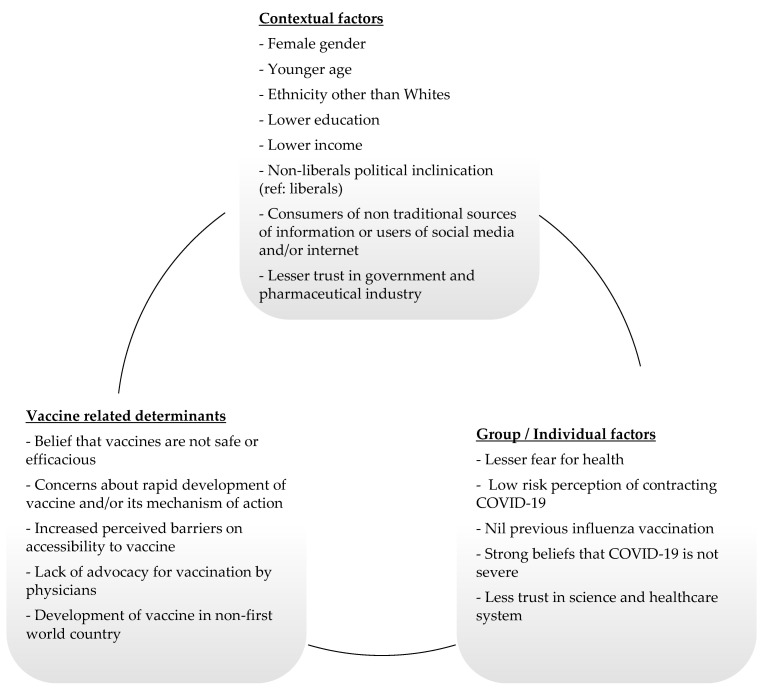
Framework diagram summarizing key determinants of vaccine hesitancy in high-income countries or regions.

**Table 1 vaccines-09-00900-t001:** Characteristics of included studies (*n* = 97).

Variables	*n* (%)
**Year of study**	
2019	0 (0)
2020	48 (49.5)
2021	49 (50.5)
**Continent of study**	
North America	42 (43.3)
Europe	33 (34)
Asia	11 (11.3)
Oceania	7 (7.2)
Cross-continents	4 (4.1)
**Country of study**	
USA	39 (40.2)
Italy	9 (9.3)
Multiple countries	9 (9.3)
Australia	6 (6.2)
France	6 (6.2)
Hong Kong	4 (4.1)
U.K.	8 (8.2)
Germany	2 (2.1)
Poland	2 (2.1)
UAE	2 (2.1)
Canada	2 (2.1)
Other countries ^a^	8 (8.2)
**Patient populations**	
General public	71 (73.2)
Healthcare workers	13 (13.4)
University students and/or university staff	5 (5.2)
Patients with autoimmune conditions	3 (3.1)
Patients with malignancy	1 (1)
Adolescents and/or children	1 (1)
Others ^b^	3 (3.1)
**Number of study participants**	
0–1000	28 (28.9)
1001–2000	32 (33)
2001–5000	25 (25.8)
5001–10,000	11 (11.3)
>10,000	1 (1)
**Study design**	
Cross-sectional study	75 (77.3)
Mixed methods	14 (14.4)
Randomized controlled trials	2 (2.1)
Pure qualitative study	1 (1)
Case control study	2 (2.1)
Longitudinal study	3 (3.1)
**Methodology of data collection ^#^**	
Online survey	87 (89.7)
Telephone interview	7 (7.2)
Paper questionnaire	3 (3.1)
Face-to-face survey	2 (2.1)
Focus group discussion	2 (2.1)
Combinations of methods ^c^	5 (5.2)
**Response rates**	
0–<25%	5 (5.2)
25–<50%	7 (7.2)
50–<75%	15 (15.5)
75–100%	17 (17.5)
Not specified	53 (54.6)
**Use of validated questionnaire**	
Yes	26 (26.8)
No	71 (73.2)
**COVID-19 vaccine hesitancy 30% or more across continents ^d,e^**	
North America	25/42 (59.5)
Europe	11/33 (33.3)
Asia	8/11 (72.7)
Oceania	2/7 (28.6)

^a^ Finland (*n* = 1), Greece (*n* = 1), Israel (*n* = 1), Japan (*n* = 1), Kuwait (*n* = 1), New Zealand (*n* = 1), Qatar (*n* = 1), Saudi Arabia (*n* = 1). ^b^ Parents/caregivers (*n* = 1), firefighters / first responders (*n* = 1), Blacks living with HIV (*n* = 1). ^c^ Studies using a combination of either of online, face to face, paper questionnaire, telephone or mail methods included. ^d^ Studies involving participants from multi-continents (*n* = 4) are omitted in the table. ^e^ Studies reporting guardians’ vaccine hesitancy for their wards included in the analysis (if a study reports both guardians’ hesitancy for wards and for themselves, the higher of the 2 is taken into consideration for count here). ^#^ Each category has been reported as per described in individual study and the net total will be more than 97 due to some studies having combination methods.

**Table 2 vaccines-09-00900-t002:** Contextual determinants of vaccine related hesitancy in high-income countries or regions.

Factor	Number of Supporting Studies	References	Number of Studies Which Found no Significance	References	Number of Disagreeing Studies	References
**Communication and media environment**						
Participants whose sources of information are mainly via social media/internet	6	[28,29,30,31,32,33]	1	[34]	0	
Lack of widely accessible information on vaccine related information	5	[29,35,36,37,38]	1	[39]	0	
Participants who are mainly users of non-traditional media (not radio, not newspapers, not television)	3	[30,32,34]	1	[33]	0	
**Influential leaders, gatekeepers and anti or pro-vaccination lobbies**						
Preferences for Donald Trump endorsements of vaccination	2	[40,41]	1	[42]	0	
Preferences for Dr. Anthony Fauci pro-vaccination endorsements	0		0		2	[41,42]
**Policies/politics**						
Political inclination: U.S. Democrats supporters	0		0		8	[33,40,41,43,44,45,46,47]
Political inclination: Non liberals (Far right, far left, conservative)	8	[40,46,48,49,50,51,52,53]	2	[22,54]	0	
Mandatory vaccination	2	[55,56]	0		0	
Political inclination: Vote abstinence	1	[48]	0		0	
**Religion**						
Increased religiosity	5	[31,37,40,56,57]	2	[41,43]	1	[30]
**Sociodemographic related**						
Females	37	[15,20,24,33,35,36,37,40,41,44,46,48,50,51,52,53,54,56,57,58,59,60,61,62,63,64,65,66,67,68,69,70,71,72,73,74,75]	12	[22,34,38,43,76,77,78,79,80,81,82,83]	1	[84]
Younger participants	31	[32,35,36,37,41,46,49,53,54,56,60,63,65,66,67,68,70,71,72,73,76,78,79,83,84,85,86,87,88,89]	12	[15,34,50,52,58,61,74,77,81,90,91,92]	10	[20,30,31,43,48,57,59,75,93,94]
Non-whites	24	[23,31,32,37,40,41,45,47,49,51,52,54,64,65,68,70,71,72,78,86,87,89,93]	6	[22,33,50,79,84,90]	1	[43]
Lower education (below college)	19	[32,34,36,41,49,54,65,66,68,69,71,77,82,85,86,89,95,96,97]	14	[20,30,33,38,43,53,56,58,76,78,79,90,92,98]	0	
Lower income	13	[23,30,40,44,48,49,51,52,53,54,82,89,97]	12	[22,33,50,56,58,68,77,80,81,86,90,92]	0	
HCW without clinical roles (ref: HCW with clinical roles)	7	[15,61,64,65,70,73,87]	0		2	[67,72]
Students in HC discipline (ref: students in non-HC disciplines)	0		1	[99]	0	
Non-Asians	4	[66,70,71,89]	3	[65,79,90]	1	[87]
Presence of child at home	3	[15,44,54]	4	[30,58,84,98]	0	
Married	2	[54,78]	5	[33,38,51,92,98]	3	[20,75,76]
**Geographical barriers (i.e., accessibility)**						
Rural regions (residence, place of practice)	5	[46,65,68,86,100]	3	[38,51,101]	1	[57]
**Pharmaceutical/governmental motives**						
Lower trust in pharmaceutical industry	4	[37,41,92,102]	0		0	
Lower trust in government	3	[20,56,79]	0		0	
**Others**						
Increased passage of time longitudinally in pandemic	5	[34,68,71,75,94]	2	[56,82]	0	
Participants without healthcare insurance	3	[52,68,84]	0		0	

Abbreviation: HC—healthcare.

**Table 3 vaccines-09-00900-t003:** Individual/group determinants of increased vaccine hesitancy in high-income countries or regions.

Factor	Number of Supporting Studies	References	Number of Studies Which Found no Significance	References	Number of Disagreeing Studies	References
**Experience with past vaccination**						
History of influenza vaccination	0		1	[103]	28	[15,19,20,22,25,33,35,36,37,41,46,47,49,50,58,61,65,66,68,73,74,75,78,79,86,89,90,95,104]
Having children with up-to-date vaccinations	0		0		1	[25]
**Beliefs, attitudes about health and prevention**						
Lesser fear for health or worry about COVID-19	16	[32,33,35,37,38,40,49,53,54,59,61,67,68,79,81,90]	2	[80,103]	2	[28,37]
Perception of lower risk of contracting COVID-19	15	[28,43,44,49,52,54,59,61,63,81,94,96,98,104,105]	4	[51,55,74,92]	1	[69]
Belief that COVID-19 is not severe	12	[22,36,41,49,52,58,63,64,81,92,98,105]	1	[43]	1	[69]
Greater conspiracy beliefs	8	[30,32,50,54,98,106,107,108]	1	[80]	0	
Belief in greater efficacy of complementary alternative medicine or one’s natural immune system	5	[20,29,62,73,102]	0		0	
Belief that COVID-19 is not a disease	4	[20,58,85,109]	0		0	
Belief that threat of COVID-19 is exaggerated	2	[85,109]	0		0	
Lesser compliance with COVID-19 prevention behaviors	2	[22,51]	0		0	
**Knowledge and awareness**						
Lower knowledge about COVID-19	4	[38,79,85,98]	3	[22,37,52]	0	
Lower knowledge about vaccination	1	[33]	2	[22,37]	0	
**Health-system and providers—trust and personal experience**						
Lesser trust in healthcare system	11	[22,30,32,50,62,68,73,92,106,107,108]	2	[80,81]	0	
Lesser trust in science or in scientist	9	[30,33,35,40,54,55,58,73,91]	0		0	
**Immunization as a social norm vs. not needed/harmful**						
Belief that vaccination is non-beneficial and/or unimportant	12	[20,31,37,43,54,68,78,90,91,92,102,105]	0		0	
Belief that vaccination is a hoax/harmful	1	(32, 92)	1	[20]	0	
**Humanistic traits**						
Lesser sense of collective responsibility e.g., protect loved ones, neighbors	10	[30,35,37,49,54,89,90,94,101,109]	2	[55,103]	0	
Lower self-efficacy	4	[31,43,54,105]	0		0	
**Other factors**						
No concomitant chronic diseases or not taking regular medications	11	[20,30,33,36,57,62,74,78,82,89,98]	8	[25,33,46,61,75,84,90,92]	2	[64,69]
Peers or family with previous COVID-19 infection	0		4	[22,30,46,103]	3	[33,67,89]
Greater desire to return to normalcy	0		0		2	[35,37]
Previously tested for COVID-19 antibodies or do not mind testing for COVID-19 antibodies	0		0		3	[40,54,65]

**Table 4 vaccines-09-00900-t004:** Vaccine related determinants of vaccine hesitancy in high-income countries or regions.

Factor	Number of Supporting Studies	References	Number of Studies Which Found no Significance	References	Number of Disagreeing Studies	References
**Risk/benefit (scientific evidence)**						
Belief that the COVID-19 vaccines are unsafe or ineffective	24	[20,22,29,35,36,37,40,41,43,50,52,54,55,63,68,73,88,90,94,102,103,108,110,111]	1	[69]	0	
Perceived duration of protection from COVID-19 vaccines to be short (one year or less)	1	[73]	0		0	
**Introduction of new vaccine/formulation**						
Concerns about rapid development, novelty, and/or mechanism of action of vaccine	9	[20,22,24,41,43,50,54,108,110]	1	[22]	0	
**Mode of administration**						
Fear of needles as a route of vaccine administration	2	[37,111]	0		0	
**Reliability or source of vaccine supply**						
Vaccines developed by first world regions (US and European Union)	0		0		3	[22,36,110]
**Vaccination schedule**						
Concerns about vaccine requiring more than one dose	2	[88,110]	0		0	
**Design of vaccination program/mode of delivery**						
Presence of perceived barriers to accessibility of vaccine (i.e., location for vaccination, time spent on transport)	4	[36,92,97,111]	0		0	
**Role of Healthcare professional**						
Lack of advocacy for COVID-19 vaccination from attending physician	3	[49,52,88]	1	[34]	0	
**Costs**						
Concerns for costs of COVID-19 vaccination	2	[97,104]	2	[43,90]	0	
Availability of monetary incentives to get vaccinated	0		1	[39]	0

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
