# Peer review of "COVID-19 Vaccine Hesitancy—A Scoping Review of Literature in High-Income Countries"

_vaccines, 2021, doi:10.3390/vaccines9080900_

Round 1
Reviewer 1 Report
The review manuscript by Aw et al. describes the COVID-19 vaccine hesitancy in some high income countries, classified by World Bank; GNI-based classification). Detailed scope and shortfalls were presented, as well as presenting some of the key findings in studies that were already published. In my opinion, the manuscript requires major editing including better organization/structure to improve readability. Some of the more detailed comments are as following:
- Authors fail to provide information on cohort size. Additional information on literacy rate may help to provide association to misinformation on vaccine.
- Authors must carefully comb through the manuscript for grammatical/typological errors and text ambiguity. Here are just some (just too many to list all):
- Line 15-36: Check font size
- Line 24: Try to be consistent with abbreviations (e.g. COVID-19 cf. COVID)
- Line 33: Please clarify what is considered high. What % or range is deemed high?
- Try to be consistent with USD or US$ (cf. USD$)
- many more...
- Figure 2: provide a better figure legend to describe the figure. The "Ns" do not add up in each region. Information missing from South America, Africa, Oceanic (no information on New Zealand?). Suggestion: Tabulate on a chart instead? World map does not add much value to the data.
- Table 2: Not comprehensible to any reader. Perhaps find a better way to structure these.
- Line 380-422: Taken these into consideration, I strongly believe that it is in authors' best interest to put forward only findings of the highest quality (line 412) and that the conclusion drawn by the authors must be
- Given that way the authors structured the conclusion (by geographical region; with a sole focus on Asia), it would make sense that all figures and tables to be categorized into the respective geographical region.
Author Response
Please see the attached MS word file if the text below has error due to formatting.
Manuscript ID: vaccines-1277111
Title: COVID-19 Vaccine Hesitancy – A scoping review of literature in High-Income Countries
Revision 1
On behalf of my co-authors, I would like to thank you very much for the careful review of the above manuscript to make it a better one
We hope to address the comments raised in the review and have included all the reviewers’ suggestions in the enclosed revised manuscript.
Reviewer #1
The review manuscript by Aw et al. describes the COVID-19 vaccine hesitancy in some high income countries, classified by World Bank; GNI-based classification). Detailed scope and shortfalls were presented, as well as presenting some of the key findings in studies that were already published. In my opinion, the manuscript requires major editing including better organization/structure to improve readability. Some of the more detailed comments are as following:
- Authors fail to provide information on cohort size. Additional information on literacy rate may help to provide association to misinformation on vaccine.
Author’s reply:
We thank you for your comment and suggestion.
Our team acknowledges the importance of such data and has adopted your suggestion. We will make available our raw data on the descriptions of individual studies included in the review in Supplementary file table S2. In it, the descriptions on the author, year of publication, reference cited linked to the manuscript, sample size, study design and methodology, age, gender, education level and participants type recruited by individual evidence sources will be displayed.
We would like to clarify on the second point on literacy rate and check if the question is referring to health literacy rate. If so, we are unable to provide data on it as the studies included in the scoping review did not report information on health literacy rate. On reflection, this will make the case strong for a future study on health literacy insights and behavioural outcomes with regards to the COVID-19 pandemic.[1]
Nonetheless, we do have the data on education level of the participants in individual evidence sources and will display the information in Supplementary file Table S2.
Due to heterogeneity in the reporting of education levels as well as a significant number of studies which did not report education level of participants (n=38), we did not further evaluate the relationship between education level and misinformation of vaccine. Nonetheless, we agree that this is an important topic which should be addressed in another study.
We have included a section discussing this limitation under the discussion section in lines 441 – 445.
1. Authors must carefully comb through the manuscript for grammatical/typological errors and text ambiguity. Here are just some (just too many to list all):
-
- Line 15-36: Check font size
Author’s reply: We would like to apologize for the typographical errors and have amended the font size as suggested.
- Line 24: Try to be consistent with abbreviations (e.g. COVID-19 cf. COVID)
Author’s reply: Thank you for the suggestion. All abbreviations have since been standardized in the revised manuscript.
- Line 33: Please clarify what is considered high. What % or range is deemed high?
Author’s reply: Thank you for your comment. We agree with you that the phrase ‘’Vaccine hesitancy in high-income countries is high’’ is ambiguous and does not add value to the manuscript. Line 25-26 has already described what is needed. We have therefore deleted the above-mentioned phrase in the manuscript.
- Try to be consistent with USD or US$ (cf. USD$)
Author’s reply: Thank you for your suggestion. We have changed the wordings uniformly to US$
- many more...
Author’s reply: We sincerely apologize for our oversight in the grammatical and typographical errors. The authors have come together to screen through the manuscript together and proofread each other. Text ambiguity and grammatical mistakes have been corrected or rephrased to give better clarity of sentences.
2. Figure 2: provide a better figure legend to describe the figure. The "Ns" do not add up in each region. Information missing from South America, Africa, Oceanic (no information on New Zealand?). Suggestion: Tabulate on a chart instead? World map does not add much value to the data.
Author’s reply:
Thank you for your comments and suggestions. The net total n=97 for included studies. Studies done in continents North America, Europe, Oceania and studies done in multi-continents (n=4) add up to 97.
During the selection process of articles in our review, we did not find any study evaluating COVID-19 vaccine hesitancy and its determinants published in high-income countries in South America and Africa. As such, no data was presented on the two continents.
3. Table 2: Not comprehensible to any reader. Perhaps find a better way to structure these.
Author’s reply:
Thank you for your comment and suggestion.
Tables 2, 3 and 4 are structured and adapted from the recommendations of the SAGE working group on vaccine hesitancy determinants matrix tables for ease of comparison and for advancement of vaccine hesitancy research.[2] We have grouped the described determinants in individual studies according to the matrix and inserted citations of the supporting and/or disproving references next to the themes along with the net count of the number of studies supporting and/or disproving the same theme.
My co-authors and I had considered using the traditional tables displayed in SRMA articles when formulating the write-up of our article where the titles and authors of the studies were spelt out in the left most column with the descriptions of the study in the same row. However, we had felt that the SAGE vaccine hesitancy determinants matrix would allow our readers to focus better on the determinants summarized.
Moreover, we recognized that there would be a substantial amount of text in the table if we were to use the traditional table format. To prevent overcrowding of the table, we have decided to use the reference number attached to the cited study instead of spelling out the author and title of the study.
The citations were provided to facilitate ease of referencing for readers. It is also hoped that this will facilitate other detailed reviews to be performed easily in the future in the field of evaluating factors associated with COVID-19 vaccine hesitancy.
The numbers of supporting studies and disproving studies in the sub-categories were provided to illustrate how well-studied the determinants were. It is hoped that this will help researchers in the field to identify factors which are less well-studied and consider including them in their research.
Lastly, we inserted Fig. 3 as an overarching summary of the findings in Tables 2, 3 and 4 to facilitate the readers’ understanding of the summarized key determinants which were most frequently studied.
We hope that our rationale for formatting the tables as above will be acceptable to you.
4. Line 380-422: Taken these into consideration, I strongly believe that it is in authors' best interest to put forward only findings of the highest quality (line 412) and that the conclusion drawn by the authors must be…
Author’s reply:
Thank you for your valuable comment and suggestion. We have considered and adopted your suggestion by including a supplementary table S2 file on the characteristic of the studies included in the review. Although not S2 is not a full critical appraisal, it will provide readers with insight into the validity of the included studies. We have also made major revisions to the discussion section of the manuscript to reflect our thoughts on the validity of included studies.
The team had deliberated a few months ago between performing a scoping review or systematic review in the initial phase of this review where the latter would have a greater emphasis on critical appraisal of the literature to be included.
After considering a key paper by Zachary Munn et al,[3] the team opted to pursue a scoping review as one of the chief aims of this review was to identify and map out the available evidence given the massive volume of individual studies across the various countries. In addition, we also recognized that some of our secondary aims for this review tied in closer to the indications for performing a scoping review. They include the following matters related to COVID-19 vaccine hesitancy in high-income countries:
- To identify the types of available evidence in a given field
- To clarify key concepts/ definitions in the literature
- To examine how research is conducted on a certain topic or field
- To identify key characteristics or factors related to a concept
- To identify and analyse knowledge gaps
Given that the main role of a scoping review is to provide an overview of the evidence rather than to produce a critically appraised / synthesized result for a particular question (as seen for a systematic review), assessment of the methodological limitations is generally not performed, as per Zachary Munn et al.
The team recognizes the importance of providing a critical appraisal of the literature and have listed this specific limitation (inherent in a scoping review) within the limitation section of our discussion. Despite this, there are multiple benefits associated with performing a scoping review as it often serves as a precursor to other systematic reviews. One of main benefits is that it enables authors to identify the nature of a broad field of evidence so that ensuing reviews can be assured of locating adequate numbers of relevant studies for inclusion.
- Given that way the authors structured the conclusion (by geographical region; with a sole focus on Asia), it would make sense that all figures and tables to be categorized into the respective geographical region.
Author’s reply:
Thank you for your suggestion. The team did not intentionally focus on Asia as this review aimed to evaluate vaccine hesitancy and its determinants in high-income regions or countries. Table 1 illustrated that Asia had 8 out of 11 studies (72.7%) that described a high vaccine hesitancy which we defined as 30% or more. North America came close at 57.1% of studies conducted showing vaccine hesitancy of 30% or more while Europe stood at only 36.4%. To minimize excessive repeating of information already reported in the results, the team shortened the conclusion which may have led to the misconception that there was a focus on Asia.
We have removed the geographical segregation component in the conclusion to avoid this potential misunderstanding. The new phrase now reads “Overall, COVID-19 vaccine hesitancy remains a highly prevalent problem in high income countries or regions” in lines 461-462 to reflect the inclusiveness of our review.
Once again, allow us to thank you for your valued time in making our paper a better one.
References
- Abel, T. and D. McQueen, Critical health literacy and the COVID-19 crisis. Health Promot Int, 2020. 35(6): p. 1612-1613.
- World Health Organisation, Report of the SAGE Working Group on Vaccine Hesitancy. 2014.
- Munn, Z., et al., Systematic review or scoping review? Guidance for authors when choosing between a systematic or scoping review approach. BMC Med Res Methodol, 2018. 18(1): p. 143.
Reviewer 2 Report
In this study, authors conducted a scoping review to summarize COVID-19 Vaccine Hesitancy and its associated factors in high-income countries. The article is of great significant and useful for the prevention of the COVID-19, especially for formulating health policies by the policymakers. However, there are some questions should be addressed before consideration.
- In the first sentence of the section of Introduction, the number of cases and deaths from COVID-19 is updated in real time, deadlines should be indicated.
- In Figure 1. Flowchart for retrieval of articles, from Screening to Eligibility, there is no more specific description in Record exclude, please provide a brief supplement.
- Also, in Figure 1, there is no number of articles "Additional Records Identified from Handsearching of" in the Final inclusion of Article. Please add.
- There are no details of the characteristics of include studies in Supplementary File 1.
- Vaccine hesitation is also significantly associated with other factors not mentioned in literature such as the category of vaccine. High-income countries have a variety of vaccines, some of which are relatively new, such as mRNA vaccine. The low level of public awareness about newly developed vaccines, whose long-term safety remains to be proven, may be an important factor in the vaccine hesitancy. If this factor is not included in the included studies, it should be discussed in the discussion section of the paper.
- The format of Table 2 and Table 3 should be the same, it is recommended to fill with “0” rather than “NA”.
- Last but not least, from my opinion, it is suggested that the word "countries" be replaced by "regions" or "countries or regions".
Author Response
Please refer to attached MS word file if there is formatting error and the text below is unreadable
Manuscript ID: vaccines-1277111
Title: COVID-19 Vaccine Hesitancy – A scoping review of literature in High-Income Countries
Revision 1
On behalf of my co-authors, I would like to thank you very much for the careful review of the above manuscript to make it a better one
We hope to address the comments raised in the review and have included all the reviewers’ suggestions in the enclosed revised manuscript.
Reviewer #2:
In this study, authors conducted a scoping review to summarize COVID-19 Vaccine Hesitancy and its associated factors in high-income countries. The article is of great significant and useful for the prevention of the COVID-19, especially for formulating health policies by the policymakers. However, there are some questions should be addressed before consideration.
1. In the first sentence of the section of Introduction, the number of cases and deaths from COVID-19 is updated in real time, deadlines should be indicated.
Author’s reply:
Thank you so much for your comment and suggestion. We wholeheartedly agree, on reflection, that it is more meaningful to attach a timeframe to the data on number of cases and deaths from COVID-19. We have decided to cite data from WHO Coronavirus (COVID-19) Dashboard instead. The amended phrase will read: ‘’Since its first reported case in December 2019, the coronavirus-2019 (COVID-19) pandemic has culminated in over 179 million infections and 3.88 million deaths globally as of 24th June 2021” in lines 46-47.
2. In Figure 1. Flowchart for retrieval of articles, from Screening to Eligibility, there is no more specific description in Record exclude, please provide a brief supplement.
Author’s reply:
Please accept our sincere apologies for the oversight due to formatting error. We have reformatted the figure and included footnotes a and b to the Figure 1 to prevent text from missing in the future. These footnotes will detail descriptions for the exclusion.
3. Also, in Figure 1, there is no number of articles "Additional Records Identified from Handsearching of" in the Final inclusion of Article. Please add.
Author’s reply:
We would like to apologise for this oversight which occurred due to formatting error. We have resized the text box and font format so that the “n” appears
4. There are no details of the characteristics of include studies in Supplementary File 1.
Author’s reply:
Thank you so much for your comment. The team will adopt your suggestion and include the data on characteristics of all studies included in our original Excel data extraction file spreadsheet in Supplementary file Table S2.
4. Vaccine hesitation is also significantly associated with other factors not mentioned in literature such as the category of vaccine. High-income countries have a variety of vaccines, some of which are relatively new, such as mRNA vaccine. The low level of public awareness about newly developed vaccines, whose long-term safety remains to be proven, may be an important factor in the vaccine hesitancy. If this factor is not included in the included studies, it should be discussed in the discussion section of the paper.
Author’s reply:
We thank you for your valuable inputs and suggestion. The fear of side effects of a vaccine developed with new technology has been explored in some included studies in our review. Table 4 on vaccine related determinants of vaccine hesitancy has a subgroup on “Concerns about rapid development, novelty, and mechanism of action of vaccine” where 9 studies found an association with this concern and vaccine hesitancy.
Furthermore, we have also adopted your suggestion to further add this into the discussion section in Line 351 – 354 with new reference cited.
The format of Table 2 and Table 3 should be the same, it is recommended to fill with “0” rather than “NA”.
Author’s reply:
Thank you very much for your comments. We will proceed to fill up the table with ‘’0’’ in tables 2, 3 and 4 in the applicable places rather than leaving it blank.
5. Last but not least, from my opinion, it is suggested that the word "countries" be replaced by "regions" or "countries or regions".
Author’s reply:
The team acknowledges that in the interconnected world we are living in, identifying by regions will be more applicable and inclusive. We have since changed the terms accordingly as per your suggestion.
Once again, allow us to thank you for your valued time in making our paper a better one.
Reviewer 3 Report
The authors undertook a high quality search of the available published literature but the analysis, discussion and presentation of their findings is problematic and lacks basic information to contextualize this work. The list below highlights a range of the issues but is not comprehensive for reasons of time and brevity.
- The Suppl table outlining which references refer to which country needs to be expanded to give details on the studies in multiple countries (which countries, n values for each country), also to give any statistical information which is referred to in the main text (e.g. Table 2 and Table 3, etc)
- Definitions of terms used across studies are required. As an example – what is ‘lower income’? Is it USD threshold or is it a socioeconomic demographic such as the bottom quartile or quintile?
- 19/25 themes in Table 2 have 10 or less studies, with 9 having less than 5 studies. Some of the themes seem to be very specific (e.g. Trump, Fauci, US Demoractic supports) and have low numbrs of studies.
- There does not appear to be any index of bias for the studies
- There does not appear to be any information in Table 2 on statistical strength of association and numbers of participants, e.g. n = 10,000, p <0.006. Whilst this is not a meta-analysis the current format gives equal weighting to all studies when we know that 28 have less that 1000 participants whereas 1 have >10,000. It is also not clear whether th number of participants matches with th number of responses or the number thesurveys etc were sent out to.
- The authors have separate entries for U.K., United Kingdom, England and Scotland. The United Kingdom of Great Britain and Northern Island is the full name of the United Kindgom and inclues England and Scotland. These results should be combied into a single entry.
- In Figure 2 it appears that there are only 33 studies for the US. There are two studies for Canada and the table lists North America as having 42 studies. Previously there are 39 studies in th US and one multi-country study in North America which does not ppear to b included in Figure 2.
- Methodology of data colletion in Table 1 is difficult to understand. The authors have a line item for combinations of methods which include a combination of different mthods. Hoever the other 5 options add up to 97. The listed methods adds up to 97 so it can’t include stduies with more than one method.
- When data is presented in Figure 2 from more than one study it is difficult to elieve tht independent studies go, to the single decimal point, identical prevlence rates, e.g. Canda has 2 studies by vaccine hesitancy listed simply as 7%.
- Data in Table 3 suffers from he same issues as Table 2 in terms of unclear statstics and power of studies, 10/22 themes have 5 studies or less, and unclear definitions of topics (e.g. how is ‘Belief that COVID19 is not severe’different from ‘Belief that severity of COVID19 is exaggerated’?). It is also crucial to know how’topics such as ‘Lower knowledge about vaccination or COVID19’ are assessed.
- There is little to no analysis of the ohort effects, e.g. younger people are more lkely to rely on non-traditional sources of media and who have social media as a source of information.
- The discussion promotes that selective countries vaccination rates align with the findings of this study on vaccine hesitancy rates in those countries, more as a qualitative or trend measure. However the uthor omit mentioning the UK which has, with 8 studies, the third highest number of studies and vaccination hesitancy of 9.9 – 36% but currently have ~90% of the adult population with at least one dose of a COVID vaccine. At the very least this should be discussed in terms of possible sampling bias, or is the results of communication about COVID vaccine safety and/or efficacy, or a range of other reasons.
Author Response
Manuscript ID: vaccines-1277111
Title: COVID-19 Vaccine Hesitancy – A scoping review of literature in High-Income Countries
Please refer to attached PDF file below if the following text below has formatting errors.
Revision 3
On behalf of my co-authors, I would like to thank the reviewer #3 for the careful review of the above manuscript to make it a better one.
We hope to address the comments raised in the review and have included all the reviewer’s suggestions in the enclosed revised manuscript.
Any change to the manuscript can be viewed under “tracked changes”.
Reviewer #3
The authors undertook a high quality search of the available published literature but the analysis, discussion and presentation of their findings is problematic and lacks basic information to contextualize this work. The list below highlights a range of the issues but is not comprehensive for reasons of time and brevity.
- The Suppl table outlining which references refer to which country needs to be expanded to give details on the studies in multiple countries (which countries, n values for each country), also to give any statistical information which is referred to in the main text (e.g. Table 2 and Table 3, etc)
Author’s reply:
Thank you for your suggestion. We have revised Supplementary S2 and expanded details on studies done in multiple countries.
The reply regarding the statistical information is better reflected in reply to comment 5.
- Definitions of terms used across studies are required. As an example – what is ‘lower income’? Is it USD threshold or is it a socioeconomic demographic such as the bottom quartile or quintile?
Author’s reply:
Thank you for your suggestion. There is considerable heterogeneity in the way different studies define these terms. The terms were reported as per described in individual studies as a form of narrative review already delineated a priori in our methods section.
- 19/25 themes in Table 2 have 10 or less studies, with 9 having less than 5 studies. Some of the themes seem to be very specific (e.g. Trump, Fauci, US Demoractic supports) and have low numbrs of studies.
Author’s reply:
Thank you for your recommendations. We have added this in as a limitation in line 426:
“Thirdly, among the 57 themes of vaccine hesitancy found in the systematic review, 26 (45.6%) themes had less than 5 studies. The percentages in Tables 2, 3 and 4 with themes having less than 5 studies were n=9 (36%), 40.9% n=9 (40.9%) and n=8 (80%) respectively. A possible insufficient exploration of a theme in the included studies has to be taken into consideration while interpreting and contextualizing the results to in-dividual country.
We would also like to point out a preponderance of studies done in the U.S. exploring the 2 sub-categories on “policies/politics” and “influential leaders, gate-keepers and anti or pro-vaccination lobbies”. Due to geopolitical differences, generalizability of these themes may be limited”.
- There does not appear to be any index of bias for the studies
Author’s reply:
The team had deliberated a few months ago between performing a scoping review or systematic review in the initial phase of this review where the latter would have a greater emphasis on critical appraisal of the literature to be included.
After considering a key paper by Zachary Munn et al,[1] the team opted to pursue a scoping review as one of the chief aims of this review was to identify and map out the available evidence given the massive volume of individual studies across the various countries. In addition, we also recognized that some of our secondary aims for this review tied in closer to the indications for performing a scoping review. They include the following matters related to COVID-19 vaccine hesitancy in high-income countries:
- To identify the types of available evidence in a given field
- To clarify key concepts/ definitions in the literature
- To examine how research is conducted on a certain topic or field
- To identify key characteristics or factors related to a concept
- To identify and analyse knowledge gaps
Given that the main role of a scoping review is to provide an overview of the evidence rather than to produce a critically appraised / synthesized result for a particular question (as seen for a systematic review), assessment of the methodological limitations is generally not performed, as per Zachary Munn et al.
The team recognizes the importance of providing a critical appraisal of the literature and have listed this specific limitation (inherent in a scoping review) within the limitation section of our discussion. Despite this, there are multiple benefits associated with performing a scoping review as it often serves as a precursor to other systematic reviews. One of main benefits is that it enables authors to identify the nature of a broad field of evidence so that ensuing reviews can be assured of locating adequate numbers of relevant studies for inclusion.
- There does not appear to be any information in Table 2 on statistical strength of association and numbers of participants, e.g. n = 10,000, p <0.006. Whilst this is not a meta-analysis the current format gives equal weighting to all studies when we know that 28 have less that 1000 participants whereas 1 have >10,000. It is also not clear whether th number of participants matches with th number of responses or the number thesurveys etc were sent out to.
Author’s reply:
Thank you for your suggestion and comment. Response rates of individual studies had been added to Supplementary Table S2.
We agree that the provision of statistical strength of association would aid in greater understanding of the impact of specific factors on vaccine hesitancy. However, there were multiple factors that limit our ability to perform the above tasks.
Firstly, there were significant heterogeneity in the assessment of the individual factors associated with increased vaccine hesitancy due to different tools, instruments and methodologies used. This in turn limits our ability to perform any meta-analysis. In view of these expected heterogeneity a priori to the design of this scoping review, a narrative review of the studies factors was performed as stated in our review paper’s methods section. Nonetheless, we believe as results of newer studies continue to emerge, a formal meta-analysis of important factors should be performed to evaluate the strength of these associations.
The second reason relates to the role of the scoping review. In general, scoping reviews do not aim to produce a critically appraised and synthesized result/answer to a particular question. Instead, the aim is to produce an overview or map of the evidence.
Nonetheless we believe that the reviewer’s point has significant research value and have added this limitation in our discussion line 446
“Lastly, assessments of methodological quality of the included studies, statistical power of determinants associated with vaccine hesitancy and meta-analysis of the vaccine hesitancy rates were not performed as these were not the primary aims of this scoping review. Moreover, the heterogeneity in the definition and assessment of vaccine hesitancy and its associated determinants in different studies would not have allowed a meaningful comparison or meta-analysis. Nonetheless, with availability of more data from future studies, this scoping review can function as a precursor to guide and direct future research effort”.
We hope that this will be acceptable to the reviewer.
- The authors have separate entries for U.K., United Kingdom, England and Scotland. The United Kingdom of Great Britain and Northern Island is the full name of the United Kindgom and inclues England and Scotland. These results should be combied into a single entry.
Author’s reply:
Thank you for your suggestion. We had combined these results in a single entry in Table 1 and in Fig 2.
- In Figure 2 it appears that there are only 33 studies for the US. There are two studies for Canada and the table lists North America as having 42 studies. Previously there are 39 studies in th US and one multi-country study in North America which does not ppear to b included in Figure 2.
Author’s reply:
Thank you for your comments. We had included only studies which reported vaccine hesitancy rates. Out of 39 studies, only 33 U.S. studies reported vaccine hesitancy rates.
- Methodology of data colletion in Table 1 is difficult to understand. The authors have a line item for combinations of methods which include a combination of different mthods. Hoever the other 5 options add up to 97. The listed methods adds up to 97 so it can’t include stduies with more than one method.
Author’s reply:
We apologized for the typo error. The online survey section should correctly reflect n=87 and had been amended. The net count for the studies with respect to methodology of data collection in Table 1 will be more than 97 as some of the included studies of this review will have a combination of methods for data collection.
- When data is presented in Figure 2 from more than one study it is difficult to elieve tht independent studies go, to the single decimal point, identical prevlence rates, e.g. Canda has 2 studies by vaccine hesitancy listed simply as 7%.
Author’s reply:
The other study by Benham et al “Attitudes, current behaviours and barriers to public health measures that reduce COVID-19 transmission…..” is a qualitative study with focus group discussion and hence does not have reportable vaccine hesitancy prevalence rate.
- Data in Table 3 suffers from he same issues as Table 2 in terms of unclear statistics and power of studies, 10/22 themes have 5 studies or less, and unclear definitions of topics (e.g. how is ‘Belief that COVID19 is not severe’ different from ‘Belief that severity of COVID19 is exaggerated’?). It is also crucial to know how ’topics such as ‘Lower knowledge about vaccination or COVID19’ are assessed.
Author’s reply:
Thank you for your comment. We have relooked at the articles pertaining to belief about COVID-19 situation. To minimize ambiguity, we have rephrased the theme “belief that severity of COVID-19 is exaggerated” to “belief that the threat of COVID-19 is exaggerated”.
We agree with the reviewer that the statistical power of studies is important. However, this review was intended as a scoping review to map the literature and provide an overview of the available research related to COVID-19 vaccine hesitancy.
We recognize the importance of the research gap that the reviewer highlighted and have included this in our limitation for future research to address in line 444
“Lastly, assessments of methodological quality of the included studies, statistical power of determinants associated with vaccine hesitancy and meta-analysis of the vaccine hesitancy rates were not performed as these were not the primary aims of this scoping review. Moreover, the heterogeneity in the definition and assessment of vaccine hesitancy and its associated determinants in different studies would not have allowed a meaningful comparison or meta-analysis. Nonetheless, with availability of more data from future studies, this scoping review can function as a precursor to guide and direct future research efforts.
Researchers who are planning to investigate COVID-19 vaccine hesitancy may want to consider adopting the standardized definition of vaccine hesitancy from SAGE workgroup in future studies and adopt standardized tools for measurement of its determinants. This will facilitate and enable future systematic reviews and meta-analyses to evaluate the variation in vaccine hesitancy rates across countries or regions as well as the temporal variation in vaccine related hesitancy and its determinants”.
We also recognize the importance of how “lower knowledge about vaccination and COVID-19” was measured. During our review, we noted that there were studies which assessed COVID-19 related knowledge and vaccination using self-designed questionnaires tailored to their population. While we share the same sentiments with the reviewer that it is an important topic, it is not the objective of this review to evaluate how this knowledge was assessed. We have included this point as a knowledge gap for future research to pursue in line 369:
“While it is not within the scope of this review, the way different themes are being measured such as knowledge about COVID-19 disease and vaccination, is an important area of research impacting on the study of vaccine hesitancy across different populations. Our review noted that most studies used self-designed instruments in the eval-uation of COVID-19 vaccination knowledge which limits cross-comparison of knowledge levels across populations. Future research should consider developing a standardized instrument for the assessment of knowledge of COVID-19 vaccine and disease which can potentially be adapted for future pandemics”.
Regarding the limited number of studies behind certain themes, we had also added in the limitation under discussion line 437:
“Thirdly, among the 57 themes of vaccine hesitancy found in the systematic review, 26 (45.6%) themes had less than 5 studies. The percentages in Tables 2, 3 and 4 with themes having less than 5 studies were n=9 (36%), 40.9% n=9 (40.9%) and n=8 (80%) respectively. A possible insufficient exploration of a theme in the included studies has to be taken into consideration while interpreting and contextualizing the results to individual country”.
- There is little to no analysis of the cohort effects, e.g. younger people are more likely to rely on non-traditional sources of media and who have social media as a source of information.
Author’s reply:
Thank you for your suggestion. We had included a possible explanation on the association between younger people and reliance on non-traditional sources of media and social media in line 352 under “Discussion”. We did not analyse this association in the “Results” section due to lack of further exploration by studies included in the review.
- The discussion promotes that selective countries vaccination rates align with the findings of this study on vaccine hesitancy rates in those countries, more as a qualitative or trend measure. However the author omit mentioning the UK which has, with 8 studies, the third highest number of studies and vaccination hesitancy of 9.9 – 36% but currently have ~90% of the adult population with at least one dose of a COVID vaccine. At the very least this should be discussed in terms of possible sampling bias, or is the results of communication about COVID vaccine safety and/or efficacy, or a range of other reasons.
Author’s reply:
Thank you for your comments. My co-authors and I have specifically mentioned UAE, US, Hong Kong and Italy in detail due to the high vaccine hesitancy summarized in our review. We were curious to correlate this to real vaccine uptake in the early part when vaccines were made available globally.
In line 381, my co-authors and I had already made provisions for real world impact of vaccine hesitancy on true uptake of COVID-19 vaccination:
“With the ongoing vaccination drive globally and evolving landscape for COVID-19, it remains premature to conclude the real-world impact of vaccine hesitancy on the true uptake of COVID-19 vaccination. Uptake can be affectedconfounded by ongoing com-munication and education efforts and interventions to reduce misinformation. In addition, it can also be confounded by logistic and administrative challenges in vaccine deployment, vaccine production capacity issues from manufacturers, affordability of vaccines and global allocation of vaccines in the context of limited supplies”
Such congruence (or incongruence) could be due to ongoing innovative vaccination drive/incentives (such as allowing for expanded social activities if fully vaccinated) and evolving landscape for COVID-19 (the latest being the more infectious delta variant that is circulating globally).
In fact, surveyed vaccine hesitancy rates in UK range from 9.9 – 36% which would translate into 64%- 90.1% uptake of COVID-19 vaccine, a percentage coinciding nicely with the information pointed out above.
Nonetheless, we acknowledge the wide range of vaccine hesitancy rates illustrated in each individual country as illustrated in Table 1 and in Fig 2. This is partly due to the heterogeneity in definition of vaccine hesitancy. We have highlighted this in line 314:
“Our review discovered that only slightly more than half of the studies conducted on COVID-19 vaccine hesitancy conformed to the SAGE proposed definition. In those studies which did not conform, participants who expressed being ‘unsure’ instead of rejecting the vaccine were excluded in the hesitancy rate, leading to a potential falsely reassuring low hesitancy rate”.
Once again, allow us to thank you for your valued time in making our paper a better one.
- Munn, Z., et al., Systematic review or scoping review? Guidance for authors when choosing between a systematic or scoping review approach. BMC Med Res Methodol, 2018. 18(1): p. 143.

Round 2
Reviewer 1 Report
The revised article by Aw et al. addressed most of the questions. Much appreciated. Just few minor comments:
Line 64: "et al."
Line 298: is this continuation of previous sentence (formatting) or a new paragraph?
Figure 2: Maybe increase the width of the individual bar? Remove horizontal grids. Start a new line for numbers below country name to improve readability.
Figure 3 Ethnicity: which one (non-white or white) displayed higher hesitancy?
Author Response
Please read the attached MS word file if the text below is unreadable due to formatting issue
Manuscript ID: vaccines-1277111
Title: COVID-19 Vaccine Hesitancy – A scoping review of literature in High-Income Countries
Revision 2
On behalf of my co-authors, I would like to thank you very much for the careful review of the above manuscript to make it a better one
We hope to address the comments raised in the review and have included all the reviewers’ suggestions in the enclosed revised manuscript.
Reviewer #1
The revised article by Aw et al. addressed most of the questions. Much appreciated. Just few minor comments:
- Line 64: "et al."
Author’s reply:
Thank you for your correction. We have added a period and changed the term as suggested
- Line 298: is this continuation of previous sentence (formatting) or a new paragraph?
Author’s reply:
Our apologies. We have started line 298 with a new paragraph.
- Figure 2: Maybe increase the width of the individual bar? Remove horizontal grids. Start a new line for numbers below country name to improve readability.
Author’s reply:
Thank you for your suggestion. We have re-formatted Figure 2 for better readability.
- Figure 3 Ethnicity: which one (non-white or white) displayed higher hesitancy?
Author’s reply:
Thank you for your comment. We have since changed the phrase to “Ethnicity other than whites” in Figure 3 to remove the ambiguity of the previous sentencing.
Once again, allow us to thank you for your valued time in making our paper a better one.